# Scapular Kinematics and Patterns of Scapular Dyskinesis in Rotator Cuff Tears: A Prospective Cohort Study

**DOI:** 10.3390/jcm12113841

**Published:** 2023-06-04

**Authors:** Umile Giuseppe Longo, Laura Risi Ambrogioni, Vincenzo Candela, Alessandra Berton, Daniela Lo Presti, Vincenzo Denaro

**Affiliations:** 1Research Unit of Orthopaedic and Trauma Surgery, Fondazione Policlinico Universitario Campus Bio-Medico, Via Alvaro del Portillo, 200, 00128 Roma, Italy; laura.risiambrogioni@unicampus.it (L.R.A.); v.candela@policlinicocampus.it (V.C.); a.berton@policlinicocampus.it (A.B.); denaro.cbm@gmail.com (V.D.); 2Research Unit of Orthopaedic and Trauma Surgery, Department of Medicine and Surgery, Università Campus Bio-Medico di Roma, Via Alvaro del Portillo, 21, 00128 Roma, Italy; 3Unit of Measurements and Biomedical Instrumentation, Campus Bio-Medico University, Via Alvaro del Portillo, 21, 00128, Rome, Italy; d.lopresti@unicampus.it

**Keywords:** scapula, kinematics, scapular dyskinesis, rotator cuff tears, range of motions

## Abstract

Scapular dyskinesis (SD) is a condition of loss of normal mobility or function of the scapula. SD is frequently observed in patients with other shoulder disorders, such as rotator cuff (RC) tears. This study evaluates the different presentations in clinical outcomes and range of motions (ROMs) in patients suffering from RC tears with and without SD. A total of 52 patients were enrolled, of which 32 patients with RC tears and SD (group A) and 20 patients with RC tears without SD (group B). Statistically significant differences between the groups in terms of clinical outcomes were identified. There were statistically significant differences in terms of flexion (*p* = 0.019), extension (*p* = 0.015), abduction (*p* = 0.005), and external rotation at 90° (*p* = 0.003) and at 0° (*p* = 0.025). In conclusion, this prospective study demonstrated that SD influences the clinical presentation of patients with RC tears in terms of clinical outcomes and ROMs, apart from internal rotation. Further studies will need to show whether these differences occur regardless of SD type.

## 1. Introduction

The scapula contributes to shoulder kinematics in multiple ways. From a biomechanical perspective, the scapula acts in synergy with the rotator cuff (RC) by modulating and transferring force to the glenohumeral and scapulothoracic joints allowing for movement. Furthermore, it also fulfils an essential role in stabilizing and maintaining the structural integrity of the shoulder by limiting excessive translation during three-dimensional motions [1,2,3].

Scapular dyskinesis (SD) is a condition of loss of normal mobility or function of the scapula [4]. The scapula’s impaired movement and resting position may be frequently observed in patients with shoulder defects, such as RC tears, labrum tears, impingement syndrome, and glenohumeral joint instability. When SD is present, the scapulothoracic girdle muscles are activated to counterbalance the altered scapular motion by configuring distinctive kinematic patterns [5,6,7].

According to the classification, three types of SD are defined based on the position of the scapula to the posterior thorax. Type I is defined as a posterior displacement from the posterior thorax of the lower medial angle of the scapula, Type II is when there is a posterior displacement from the posterior thorax of the entire medial edge of the scapula, Type III is when early scapular elevation or excessive/insufficient upward scapular rotation (dysrhythmia) occurs during dynamic observation, and Type IV by normal scapula [8,9,10,11].

Since SD is often associated with other shoulder pathologies, its prevalence is underestimated. Even though limited epidemiological studies are available in the literature, SD appears to be a more typical condition of overhead athletes due to the increased functional demands on the scapulothoracic girdle [12,13,14,15,16,17]. Although SD can remain clinically silent, prolonged abnormal activation of the stabilizing muscles of the scapulothoracic girdle can lead to pain. However, it has been observed that in asymptomatic patients with SD, kinematic alteration does not affect the activity of the scapulothoracic muscles. Therefore, whether SD is responsible for the onset of pain or vice versa should be critically considered.

RC tear ranks among shoulder disorders as a condition mainly associated with SD. RC tear typically presents with pain, muscle weakness, and joint motion impairment. RC tears are multifactorial conditions that may require a personalized approach ranging from conservative, minimally invasive, or surgical treatments [18]. Although many studies have investigated what is the optimal management, to date, there is no consensus. Conservative approaches involve not only physiotherapy but also other techniques, such as extracorporeal shockwave therapy and hyaluronic acid injections. A recent study has shown that the combination of these both can be more effective than their separate use. In addition, factors such as gender could influence the outcomes of these therapies, emphasizing the importance of individualized treatments [19].

This study aims to evaluate the different presentations in terms of clinical outcomes and range of motions (ROMs) of patients suffering from RC tears with and without SD. We hypothesize that in RC tear patients, the concomitant presence of SD results in alterations in ROMs.

## 2. Materials and Methods

### 2.1. Study Design

This non-intervention observational prospective study was performed at the Department of Orthopedic and Trauma Surgery at the University Campus Bio-Medico of Rome.

We conducted a prospective cohort study enrolling patients with an RC tear admitted to the outpatient department of Orthopedics for ten consecutive months, from October 2017 to July 2018.

To be eligible, patients were stratified and screened by inclusion and exclusion criteria. Inclusion criteria were tailored to enlist the broadest possible sample of patients to the study. Instead, the exclusion criteria were meant to identify the most representative sample of patients by removing the most significant number of confounders that could bias the results.

Inclusion criteria

males and females aged ≥ 18;patients with RC tear confirmed by MRI without previous shoulder surgery and other shoulder diseases (i.e., instability, frozen shoulder, fractures, inflammatory joint disease);shoulder pain, with or without limited shoulder movement and SD.

Exclusion criteria

pediatric patients;patients without RC tear;presence of shoulder pathologies ≥ 2, or previous surgery of the shoulder;patients with body surface markers affecting the assessment, such as obesity (body mass index greater than 30);patients unable to complete assessment by clinical scores and ROMs.

The principal investigator (U.G.L.) promoted the study’s objectives, content, and participation during physician visits. Informed consent for the use of data, photographs, and videos was obtained from all patients enrolled. The study was conducted according to the guidelines of the Declaration of Helsinki and approved by the Institutional Review Board of Campus Bio-Medico University of Rome (COSMO study, Protocol number: 78/18 OSS ComEt CBM, 16/10/18). The study was developed following Good Clinical Practice (GCP) guidelines.

### 2.2. Data Collection

Each patient participating in the study was scheduled for a high-resolution MRI (1.5 T) of the injured shoulder and an orthopedic examination at the University Campus Bio-Medico of Rome. The MRIs were examined by a blinded radiologist who estimated the size of RC tear in centimeters. During the orthopedic examination, a fully trained shoulder surgeon (U.G.L.) extracted the following demographic data: age (years), height (meters), weight (kilos), sex (male/female), shoulder with RC tear (left/right), arm dominance (left/right), and evaluated each patient using clinical scores and ROMs. After the examination, the patient was taken to a room free of objects where the movements of forward elevation, internal and external rotation were recorded through a fixed-camera system. The movements were recorded by setting the camera on both the posterior and lateral sides of the patient, thus producing two videos for each candidate. Two independent investigators (V.C. and L.R.A.) subsequently evaluated these videos. In case of movement performed incorrectly, partially, or excessively fast, the patient was instructed to return to the resting position and perform it again. In these cases, a new recording was made, and all videos were examined.

### 2.3. Scapular Dyskinesis Assessment

A surgeon (U.G.L.) performed a standardized physical examination on all patients. The presence of SD was evaluated using a comprehensive method that combined visual observation and palpation of the scapula during arm movements with and without weighted loads [9].

According to the classification, three types of SD are defined based on the position of the scapula to the posterior thorax. Type I is defined as a posterior displacement from the posterior thorax of the lower medial angle of the scapula, Type II is when there is a posterior displacement from the posterior thorax of the entire medial edge of the scapula, Type III is when early scapular elevation or excessive/insufficient upward scapular rotation (dysrhythmia) occurs during dynamic observation, and Type IV is characterized by normal scapular movement, defined as no evidence of abnormality in the resting position or dynamic motions [8,9,11,20]. The investigators (V.C. and L.R.A.) independently observed the videos and assigned a kinematic pattern to each patient. They repeated the pattern assignment and established their definitive personal diagnosis by reviewing the videos two weeks later. Then, the two investigators compared their definitive personal diagnoses. For patients who received the same diagnosis, that kinematic pattern was assigned. For patients who received different diagnoses, the investigators reviewed the videos together. They resolved disagreements by consensus, and a third investigator (V.D.) was consulted when necessary. The inter-rater and intra-rater reliability and agreement of the SD test among the investigators were assessed. Patients were divided into two groups following the SD assessment: patients with RC tear and SD (group A), patients with RC tear without SD (group B).

### 2.4. Functional Assessment and Range of Motion

The study participants completed the Constant and Murley score (CMS) [21], American Shoulder and Elbow Surgeons (ASES) score [22], Disability of the Arm, Shoulder and Hand (DASH) score [23,24], and Oxford Shoulder Score (OSS) [25,26].

The CMS system [21], normalized for patient age and gender, was used to evaluate preoperative and postoperative shoulder function. It evaluates both subjective and objective functions through four domains, including pain (15 points), activities of daily living (20 points), range of movement (40 points), and strength (25 points). The total score ranges from 0, indicating a person with the worst shoulder function, to 100 points, indicating an asymptomatic and healthy person. Pain was assessed with the pain score according to the system of CMS [21], ranging from 0, indicating the severest imaginable pain, to 15 points, indicating no pain.

The ASES score is composed of two sections: the first, defined as pASES, is a patient’s self-report, and the second, defined as cASES, is used by physicians in order to record the shoulder examination findings. In this study, we only considered the patient’s self-report, which is divided into 11 items subcategorized into 2 areas, pain (1 item) and ADL (activity daily living, 10 items). The pain was measured by means of a visual analogue scale (VAS) divided into 1 cm increments and anchored with verbal descriptors at 0 (no pain at all) and 10 cm (pain as bad as it can be). Every ADL has a score that ranges from 0 (unable to do) to 3 (not difficult), with a maximum score of 30 [27,28].

The DASH score [23] is a completed subjective score with 30 analogue scale responses, and it ranges from 0 (no disability) to 100 points (severe disability). This standardized questionnaire assesses the symptoms and functional status of people with different upper limb musculoskeletal disorders. It consists of three sections: the first section, composed of 30 items, evaluates symptoms and functional status at the level of disability; the second and third sections are an optional module of four items for Sport and Music and four items for Work. Each item is scored with a five points scale (1 = no difficulty; 2 = mild difficulty; 3 = moderate difficulty; 4 = severe difficulty; 5 = unable) that are summarized and transformed to define the DASH score.

The OSS was designed as a joint-specific instrument to minimize the influence of comorbidity ranging from 12 to 60 points. It is composed of 12 questions, and each of them was scored from 1 to 5, with 1 representing the best outcome/least symptoms. Scores from each question were added, so the best overall score was 12 points. This scoring system was converted to the 0–48 scoring system, where the best outcomes are represented by the 37–48 range [25,26].

The ROMs were measured with a standard universal goniometer. Patients were positioned supine on an examination table with the shoulder abducted 90° in the scapular plane (approximately 15° anterior to the coronal plane). Supine forward elevation (sagittal plane) and internal and external rotation (90° abduction) were scored using standard measurement guidelines. During the test, the examiner (A.B.) stabilized the scapula with one hand while passively assisting the shoulder to reach the position while the forearm was held in neutral rotation. After establishing a firm endpoint, two examiners maintained the shoulder position while a third examiner (U.G.L.) performed the ROM measurement. For each shoulder position, three measurements were taken. Then, the average value was determined for statistical purposes [29].

### 2.5. Statistics

Data were recorded in a database and analyzed using IBM SPSS Statistics for Windows, version 26.0 (Armonk, NY, USA; IBM Corp). Continuous variables were expressed as mean and standard deviation and categorical variables as frequencies and percentages. Descriptive statistics were performed to describe the group and distribution of the variables. To assess data normality, the Shapiro–Wilk test was used. The Independent T-test or the Independent-Samples Mann–Whitney U test was used to find statistically significant differences in scores (ASES, CMS, DASH and OSS) and ROMs between the two groups (A and B), as appropriate. The Chi-squared test was used to assess statistically significant differences in categorical variables. Two-way random-effect intra-class correlation coefficients (ICC) were calculated for the inter-rater and intra-rater reliability tests, along with their confidence intervals. An ICC value of < 0.4 was interpreted for poor reliability; 0.4–0.75 for fair to good reliability; and >0.75 for excellent reliability. *p*-values < 0.05 were considered statistically significant.

## 3. Results

A total of 52 patients met the inclusion criteria and were enrolled in this study. The SD assessment identified 32 patients with RC tears and SD (Group A) and 20 patients with RC tears without SD (Group B). Different SD patterns were stratified within Group A. In total, 4 out of 32 patients were classified as Type 1 SD, 6 as Type 2 SD, and the remaining 22 as Type 3 SD.

For SD assessment, fair to good reliability was demonstrated using visual estimation (intra-rater examinator A ICC = 0.4, 95% CI −0.8–0.69; intra-rater examinator B ICC = 0.74, 95% CI 0.5–0.93; inter-rater ICC = 0.41, 95% CI −0.83–0.62). Only 4 out of 52 patients had a different diagnosis from the two investigators, but consensus solved them.

### 3.1. Data Analysis

This study included 52 patients (32 in group A and 20 in group B). Within Group A, 15 females and 17 males with an average age of 60.4 ± 8.7 years (41 to 79 years) were considered. Namely, 2 females and 2 males were diagnosed with type 1 SD, 3 females and 3 males with type 2 SD, and 10 females and 12 males with type 3 SD. Overall, the average height and weight were 1.67 ± 0.9 m and 74.4 ± 15 kg, respectively. The average RC tear in centimeters measured at MRI was 1.77 ± 1.02 cm.

Among Group B, 7 females and 13 males with an average age of 61.7 ± 8.7 years (ranging from 66 to 77 years) were included. The mean height and weight were 1.68 ± 0.08 m and 78.4 ± 11.5 kg, respectively. Overall, the mean size of the RC tear measured at MRI was 1.60 ± 0.78 cm (Table 1).

The presence of SD is not affected by age (*p* = 0.597), height (*p* = 0.816), weight (*p* = 0.307), sex (*p* = 0.399), and preoperative tear size (*p* = 0.553).

### 3.2. Functional Assessment and Range of Motion Analysis

The OSS score, ASES score, CMS, and DASH score were recorded for each patient. Statistical analysis compared the results of Group A and Group B (Figure 1).

Statistically significant differences in ASES (*p* = 0.003), CMS (*p* = 0.033), DASH (*p* = 0.006), and OSS (*p* = 0.006) were found among the two groups (Table 2).

There were statistically significant differences between the two groups in terms of ROMs, as concerns flexion (*p* = 0.019), extension (*p* = 0.015), abduction (*p* = 0.005), and external rotation at 90° (*p* = 0.003) and at 0° (*p* = 0.025), but not internal rotation at 90° (*p* = 0.075) and 0° (*p* = 0.438). The ROM scores are summarized in Table 3 and in Figure 2.

## 4. Discussion

This non-interventional prospective cohort study that analyzed the clinical implications of SD in patients with RC tears showed statistically significant differences in flexion, extension, abduction, and external rotation. On the other hand, no statistically significant difference was identified for internal rotation movements. Statistically significant differences in all outcomes recorded (ASES, CMS, DASH, and OSS) were found among the two groups.

SD is a clinical condition characterized by a dysfunctional movement of the scapula compared to the thorax [30,31,32]. Different patterns of SD are identified depending on which region of the scapula exhibits altered biomechanics and how severe the dysrhythmia is. Type I is defined as a posterior displacement from the posterior thorax of the lower medial angle of the scapula, Type II is when there is a posterior displacement from the posterior thorax of the entire medial edge of the scapula, Type III is when early scapular elevation or excessive/insufficient upward scapular rotation occurs during dynamic observation, and Type IV is characterized by normal scapula [8,9,11]. The alterations in shoulder kinematics that often lead to SD are frequently associated with other shoulder pathologies, such as RC tears, labrum tears, impingement syndrome, and glenohumeral joint instability. However, whether SD leads to the development of these diseases or is a consequence has yet to be clarified [33].

This non-interventional observational prospective cohort study investigated the clinical implications of SD in patients with RC tears. Fifty-two patients with RC tears were enrolled, and the presence of SD was assessed using a fixed camera system. Patients with SD were placed in Group A, and those without SD (Type 4 according to the classification) in Group B. For all patients, clinical scores and ROMs were evaluated. The two groups showed statistically significant differences in CMS, DASH, ASES, and OSS scores. These scales assess disability, residual function, and quality of life in patients with upper extremity disease. The CMS is a scale that determines the residual function after shoulder treatment. The items explored are pain, activities of daily living, strength, and ROMs (forward elevation, external rotation, abduction, and internal rotation of the shoulder) [21]. The DASH score is designed of 30 questions that allow for a standardized assessment of the impact of upper extremity pathology on function. Twenty-one questions test the ability to perform activities of daily living, six questions assess specific symptoms (e.g., pain, paresthesia, sleep disturbance), and three questions assess social or occupational limitations [23]. The ASES evaluates two dimensions of shoulder function: pain and performance in activities of daily living. By assigning a score from 0 to 100, it can quantify the degree of disability from shoulder diseases [28]. The OSS is an intuitive score of 12 items with five possible answers each. The OSS was designed to assess the impact on quality of life in patients with degenerative shoulder conditions as RC tears [34]. In particular, the OSS demonstrates a lower dispersion of errors and the absence of associations with ceiling/floor effects [35]. The statistically significant difference in favor of Group A may be explained by the fact that patients with SD have more significant functional impairment than patients without SD for similar RC tears. Therefore, the finding of significant results on four clinical scales suggests a real potential negative clinical impact of SD in patients with RC tears.

This study showed statistically significant differences in flexion, extension, abduction, and external rotation. On the other hand, no statistically significant difference was identified for internal rotation movements. The scapula plays a crucial role in shoulder kinematics. From a biomechanical perspective, during shoulder motions, the humerus rotates, imparting scapular rotation on the glenoid. Thus, the synchronous rotation of the scapula above the humeral head ensures both movement and joint stability [1,2]. Therefore, an impairment of scapular kinematics can further reduce shoulder function in patients with RC tears. Studies have shown that patients with RC tears experience a decrease in upward scapular rotation, a decrease in posterior scapular tilt, and an increase in scapular elevation [36,37]. These findings have also been encountered in cohorts of overhead athletes with SD. Specifically, in overhead athletes with Type III SD, an alteration in upward scapular rotation emerged at both 45° and 90° of shoulder abduction [38]. On the contrary, other evidence suggests no change in these movements or an increase in upward scapular rotation. To date, no uniform kinematic dysfunction has been attributed in patients with RC tears and SD [39]. This lack of findings is justified by the difficulty of assessing scapular movements in different spatial planes and the absence of clinical assessment to quantify SD. Indeed, shoulder asymmetry and ROM limitation are recognizable signs of SD, but they cannot discriminate between symptomatic and asymptomatic patients and do not provide information about the quality of life and prognosis [40].

This is the first study to evaluate clinical scores and ROMs in patients with RC teas with and without SD. Unfortunately, the lack of evidence on this topic prevents a comparison of our findings with previous studies. According to the Second Consensus Conference on the Scapula (Lexington, Kentucky) [33], SD should be considered when planning treatment of RC tears patients. Therefore, the rehabilitation program for RC tears must also focus on SD treatment, given the fundamental biomechanical role that the scapula holds [33]. However, to guide the treatment of SD, it is essential to seek a correct clinical and diagnostic framing of this disease. Therefore, further studies should investigate potential clinical outcomes and patterns of ROM alterations associated with different types of SD in patients with RC tears.

SD treatment requires re-establishing scapular position as a prerequisite for a proper recovery of shoulder kinetics [11]. Both conservative and surgical approaches are available [41,42]. Conservative treatments comprise specific exercises that improve muscle flexibility to reduce scapular traction [43,44]. Among these exercises, stretching with horizontal shoulder abduction at 90° and 150° of elevation improves the status of the pectoralis minor muscle and, consequently, external rotation and posterior tilt of the scapula during forward elevation [43,44,45]. Muscle strengthening exercises also improve strength and prop [38,46,47]. Kinesio taping on the upper and lower trapezius muscles can rebalance the scapular muscles by increasing upward scapular rotation [48]. It has been reported that SD exercises with electrical stimulation, performed to 120° shoulder abduction, enhance the distance of the spine from the scapula [49]. In overhead athletes (e.g., baseball pitchers), the shoulder joint is predisposed to experience alterations in glenohumeral joint pattern, ROM deficits, and muscle weakness, leading to SD whose magnitude of impairment increases with the level of competition [50,51,52,53,54,55]. Because of the variety and rapidity of shoulder changes, overhead athletes must be constantly monitored during the competitive season [56]. In this population, treatments focusing on intensive nonsurgical approaches provide better results, and the physical training protocol for scapular muscle strengthening could be integrated into the usual daily exercises.

When the conservative approach fails, or internal joint damage occurs (e.g., AC separation, GH injury, scapular muscle detachment), surgical treatment should be considered [6,57]. When SD is caused, it is unclear whether treatment should focus on the cause or the altered kinematics. Furthermore, removal of the cause does not necessarily lead to the rebalancing of scapular kinematics, and conversely, correction of SD does not always resolve the associated shoulder pathology [11].

The strength of this study is the use of a validated clinical method for identifying SD, which showed satisfactory reliability for clinical use [58]. Moreover, the two groups were homogeneous regarding age, weight, height, and tear size. In addition, the use of multiple clinical scales allowed for greater robustness of the results. However, this study has several limitations. First, since this study was single-center and the enrollment time-limited, the number of patients included is small, and the data should be evaluated in a larger group. A larger group will confirm our findings and can stratify the results achieved in this study by the type of SD (Type I, II, III, or IV). Clinical practice and evidence from the literature suggest that SD is a condition that impacts the scapulohumeral joint kinetics by altering the range of joint motion and negatively impacting the patient’s symptoms. A limitation of the present study is the lack of analysis of patients’ muscle strength. Subsequent studies are needed to determine whether the loss of strength related to RC tear may be worsened by the co-presence of SD. Finally, due to the lack of validated diagnostic tests to define the presence of SD, it has not been feasible to conduct an analysis to assess whether a longer SD condition would impact the clinical outcomes of patients with RC tears in a statistically significant way.

## 5. Conclusions

SD influences the clinical presentation of patients with RC tears in terms of clinical outcomes and ROMs as flexion, extension, abduction, and external rotation at 90° and 0°. However, there are no statistically significant differences in internal rotation between the groups. A comprehensive assessment of shoulder kinematics in patients with RC tears is needed to understand the biomechanical and clinical role of SD and the best treatment strategy. Further studies should be designed to investigate potential clinical outcomes and patterns of ROM alterations associated with different patterns of SD in patients with RC tears.

## Figures and Tables

**Figure 1 jcm-12-03841-f001:**
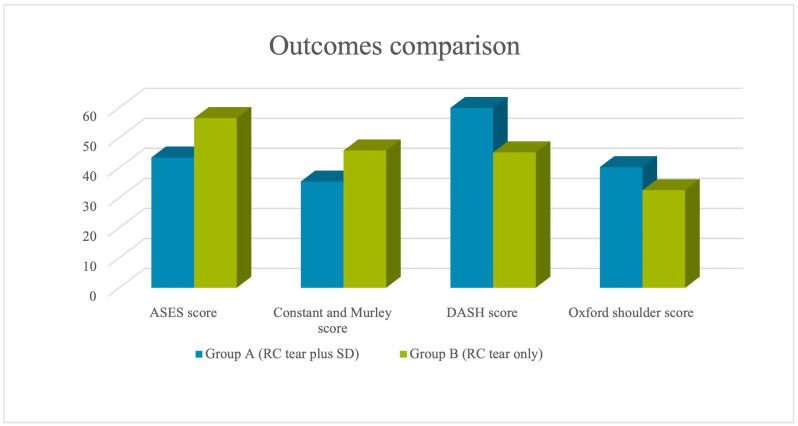
Outcomes comparison.

**Figure 2 jcm-12-03841-f002:**
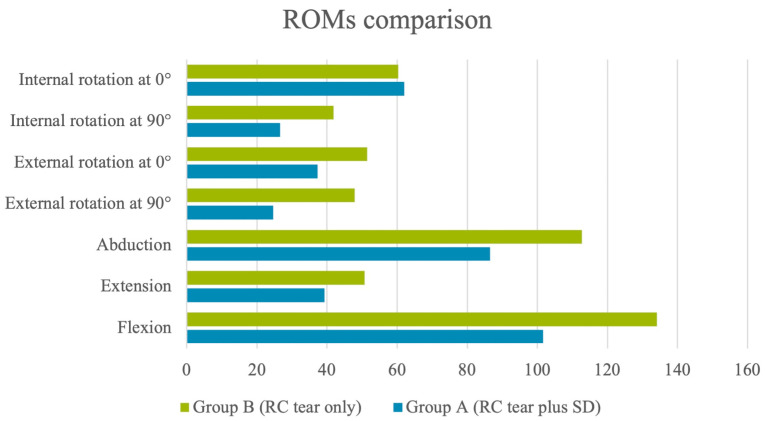
Range of motions comparison.

**Table 1 jcm-12-03841-t001:** Groups characteristics.

	Group A (*n* = 32)	Group B (*n* = 20)	*p*-Value
Age (years + SD)	60.4 ± 8.7 (41–79)	61.7 ± 8.7 (66–77)	0.597
Height (m + SD)	1.67 ± 0.9	1.68 ± 0.09	0.816
Weight (kg + SD)	74.4 ± 15.0	78.4 ± 11.5	0.307
Sex (female/male)	15/17	7/13	0.399
Tear size (cm + SD)	1.77 ± 1.02	1.60 ± 0.78	0.553

SD: standard deviation.

**Table 2 jcm-12-03841-t002:** Clinical Scores.

Group	Group A	Group B	Total	*p*-Value
	Mean ± SD	Mean ± SD	Mean ± SD	
ASES Score	43.4 ± 16.1	56.5 ± 11.2	48.5 ± 15.7	0.003 *
Constant and Murley Score	35.4 ± 15.8	45.7 ± 17.6	39.4 ± 17.1	0.033 *
DASH Score	59.9 ± 19.1	45.1 ± 16.0	54.2 ± 19.3	0.006 *
Oxford shoulder Score	40.2 ± 9.9	32.5 ± 8.5	37.2 ± 10.1	0.006 *

SD: standard deviation; * = Statistically significant (i.e., *p* < 0.05).

**Table 3 jcm-12-03841-t003:** Range of Motions.

Group	Group A	Group B	Total	*p*-Value
	Mean ± SD	Mean ± SD	Mean ± SD	
Flexion	101.7 ± 44.6	134.1 ± 24.2	114.4 ± 40.9	0.019 *
Extension	39.3 ± 12.3	50.7 ± 19.4	43.6 ± 16.2	0.015 *
Abduction	86.5 ± 31.6	112.7 ± 30.4	96.8 ± 33.4	0.005 *
External rotation at 90°	24.6 ± 23.5	47.9 ± 21.4	33.3 ± 25.2	0.003 *
External rotation at 0°	37.3 ± 22.9	51.5 ± 18.3	42.6 ± 22.2	0.025 *
Internal rotation at 90°	26.6 ± 27.8	41.9 ± 31.8	32.3 ± 30.0	0.075
Internal rotation at 0°	62.0 ± 26.1	60.3 ± 18.3	61.4 ± 23.3	0.438

SD: standard deviation; * = Statistically significant (i.e., *p* < 0.05).

## Data Availability

The datasets used and/or analyzed during the current study are available from the corresponding author on reasonable request. The access to the database is on request. All data were obtained by the Direzione Generale della Programmazione Sanitaria—Banca Dati SDO of the Italian Ministry of Health.

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
