# Peer review of "Scapular Kinematics and Patterns of Scapular Dyskinesis in Rotator Cuff Tears: A Prospective Cohort Study"

_jcm, 2023, doi:10.3390/jcm12113841_

Round 1
Reviewer 1 Report
Dear Authors
Congratulations on the study. I see that the duration of the ailment also has an impact on the clinical performance of the patients along with the strength of the rotator cuff muscles examined. If the authors could also incorporate that data into the analysis it would further strengthen the results obtained. The authors could also discuss the management principles to be followed in the treatment of either of the group for the benefit of the readers. The authors could present some visual representation of the outcomes to make it more appealing to the readers.
Minor language editing needed
Author Response
I see that the duration of the ailment also has an impact on the clinical performance of the patients along with the strength of the rotator cuff muscles examined. If the authors could also incorporate that data into the analysis it would further strengthen the results obtained. The authors could also discuss the management principles to be followed in the treatment of either of the group for the benefit of the readers. The authors could present some visual representation of the outcomes to make it more appealing to the readers.
ANSWER: We thank the reviewer for his/her suggestions.
Strength analysis has not been included in the study design since our initial assumption was that scapular dyskinesia may result in mechanical alteration to shoulder kinetics without burdening muscle strength. We hypothesized that variations in strength would be related merely to organic muscle damage, and not to the presence of altered kinematics. Thanking the author for this insight, we decided to include this lack of analysis within the limitations of the study. Indeed, we state:
“Clinical practice and evidence from the literature suggests that SD is a condition that impacts on the scapulohumeral joint kinetics by altering the ROM and negatively impacting the patient’s symptoms. A limitation of the present study is the lack of an analysis on patients’ muscle strength. Subsequent studies are needed to determine whether the loss of strength related to RC tear may be worsened by the co-presence of SD.”
The absence of tests that allow SD characterization and diagnostic underestimation make it currently impossible to estimate the onset and duration of SD. Despite the high suspicion that the longer SD persists the more shoulder kinetic patterns may be affected, in the enrolled patients it has been impossible to determine how long they had been impaired. We also added this into the limitations of the study as follows:
“Finally, due to the lack of validated diagnostic tests to define the presence of SD, it has not been feasible to conduct an analysis to assess whether a longer SD condition would impact the clinical outcomes of patients with RC tear in a statistically significant way.”
Within the discussion section, we have also implemented the management of patients with SD as follows:
“SD treatment requires re-establishing scapular position as a prerequisite for a proper recovery of shoulder kinetics [36]. Both conservative and surgical approaches are available [21, 24]. Conservative treatments comprise specific exercises that improve muscle flexibility to reduce scapular traction [56, 57]. Among these exercises, stretching with horizontal shoulder abduction at 90° and 150° of elevation improves the status of the pectoralis minor muscle and, consequently, external rotation and posterior tilt of the scapula during forward elevation [43, 56, 57]. Muscle strengthening exercises also improve strength and prop [4, 51, 54]. Kinesio taping on the upper and lower trapezius muscles can rebalance the scapular muscles by increasing upward scapular rotation [17]. It has been reported that SD exercises with electrical stimulation, per-formed to 120° shoulder abduction, enhance the distance of the spine from the scapula [58]. In overhead athletes (e.g., baseball pitchers), the shoulder joint is predisposed to experience alterations in glenohumeral joint pattern, ROM deficits, and muscle weakness, leading to SD whose magnitude of impairment increases with the level of competition [2, 20, 27, 28, 45, 53]. Because of the variety and rapidity of shoulder changes, overhead athletes must be constantly monitored during the competitive season [52]. In this population, treatments focusing on intensive nonsurgical approaches provide better results, and the physical training protocol for scapular muscle strengthening could be integrated into the usual daily exercises.
When the conservative approach fails or internal joint damage occurs (e.g., AC separation, GH injury, scapular muscle detachment), surgical treatment should be considered [15, 26]. When SD is caused, it is unclear whether treatment should focus on the cause or the altered kinematics. Furthermore, removal of the cause does not necessarily lead to rebalancing of scapular kinematics, and conversely, correction of SD does not always resolve the associated shoulder pathology [36]”
We added complementary images by extracting data from Tables 2 and 3, which summarize the statistically significant data of the study. We thank for your valuable advice that will enable us to ensure smoother reading of the study.
We hope this is now acceptable.
We hope this is acceptable.
We thank the Editorial Board for having given us the opportunity to revise our manuscript. We appreciate yours and the reviewer’s comments. I hope that the additions have now improved the manuscript, and that it has now reached the standard necessary to be formally accepted for publication in the Journal of Clinical Medicine.
Yours sincerely,
Umile Giuseppe Longo

Reviewer 2 Report
Interesting study: The aim of the study is centered on clinical description of scapular dyskinesis in rotator cuff tears.
Title: concise and comprehensive.
Abstract: The conclusions of the study are not clearly reported and are generic.
Introduction: clinical minifestations of tendinopathy, shoulder rotator cuff rupture, and conservative treatment options are not described ( see: Pellegrino, R. Journal of Sports Medicine and Physical Fitness, 2022, 62(9), pp. 1211-1218;
Pellegrino, R. BMC Musculoskeletal Disorders, 2022, 23(1), 863) as well as the study hypothesis is not described.
Study design: Specify the participant recruitment setting.
Specify and better separate inclusion and exclusion criteria.
Study sample size is missing, which should be discussed within the limitations of the study.
Discussion: the first sentence summarizing the most important findings of the paper is missing.
The description of the outcomes ( scales ) should be reported in the methods.
Author Response
- Abstract: The conclusions of the study are not clearly reported and are generic.
ANSWER: We thank the Reviewer for giving us the possibility to clarify the Abstract. We now state:
“Statistically significant difference between the groups in terms of clinical outcomes were identified. There were statistically significant differences in terms of flexion (p = 0.019), extension (p = 0.015), abduction (p = 0.005) and external rotation at 90° (p = 0.003) and at 0° (p = 0.025). In conclusion, this prospective study demonstrated that SD influences the clinical presentation of patients with RC tears in terms of clinical outcomes and ROMs, apart from internal rotation. Further studies will need to show whether these differences occur regardless of SD type.”
We hope this is now acceptable.
- Introduction: clinical manifestations of tendinopathy, shoulder rotator cuff rupture, and conservative treatment options are not described (see: Pellegrino, R. Journal of Sports Medicine and Physical Fitness, 2022, 62(9), pp. 1211-1218; Pellegrino, R. BMC Musculoskeletal Disorders, 2022, 23(1), 863) as well as the study hypothesis is not described.
ANSWER: We thank the reviewer for this suggestion. We added the suggested information as follow:
“RC tear ranks among shoulder disorders as a condition mainly associated with SD. RC tear typically presents with pain, muscle weakness, and joint motion impairment. RC tears are multifactorial conditions that may require a personalized approach ranging from conservative, minimally invasive, or surgical treatments [46]. Although many studies have investigated what is the optimal management, to date there is no consensus. Conservative approaches involve not only physiotherapy, but also other technique as extracorporeal shockwave therapy and hyaluronic acid injections. A recent study has shown that the combination of these both can be more effective than their separate use. In addition, factors such as gender could influence the outcomes of these therapies, emphasizing the importance of individualized treatments [47].
This study aims to evaluate the different presentations in terms of clinical outcomes and range of motions (ROMs) of patients suffering from RC tears with and without SD. We hypothesize.”
We have added in the bibliography the references used to enrich the introduction section.
We hope this is acceptable.
- Study design: Specify the participant recruitment setting.
ANSWER: We thank the reviewer for his/her suggestion. We modified the Study design subsection as follow:
“This non-intervention observational prospective study was performed at the Department of Orthopedic and Trauma Surgery at the University Campus Bio-Medico of Rome.
We conducted a prospective cohort study enrolling patients with RC tear, admitted to the outpatient department of Orthopedics for ten consecutive months, from October 2017 to July 2018.
To be eligible, patients were stratified screened by inclusion and exclusion criteria. Inclusion criteria were tailored to enlist the broadest possible sample of patients to the study. Instead, the exclusion criteria were meant to identify the most representative sample of patients by removing the most significant number of confounders that could bias the results.
Inclusion criteria
- males and females aged ≥ 18;
- patients with RC tear confirmed by MRI without previous shoulder surgery and other shoulder diseases (i.e., instability, frozen shoulder, fractures, inflammatory joint disease)
- shoulder pain, with or without limited shoulder movement and SD.
Exclusion criteria
- pediatric patients;
- patients without RC tear;
- presence of shoulder pathologies ≥ 2, or previous surgery of the shoulder;
- patients with body surface markers affecting the assessment, such as obesity (body mass index greater than 30);
- patients unable to complete assessment by clinical scores and ROMs.
The principal investigator (U.G.L.) promoted the study's objectives, content, and participation during physician visits. Informed consent for the use of data, photographs, and videos was obtained from all patients enrolled. The study was conducted according to the guidelines of the Declaration of Helsinki and approved by the Institutional Review Board of Campus Bio-Medico University of Rome (COSMO study, Protocol number: 78/18 OSS ComEt CBM, 16/10/18). The study was developed following Good Clinical Practice (GCP) guidelines.”
We hope this is now acceptable.
- Specify and better separate inclusion and exclusion criteria.
ANSWER: We thank the reviewer for this suggestion. We now state:
“Inclusion criteria were tailored to enlist the broadest possible sample of patients to the study. Instead, the exclusion criteria were meant to identify the most representative sample of patients by removing the most significant number of confounders that could bias the results.
Inclusion criteria
- males and females aged ≥ 18;
- patients with RC tear confirmed by MRI without previous shoulder surgery and other shoulder diseases (i.e., instability, frozen shoulder, fractures, inflammatory joint disease)
- shoulder pain, with or without limited shoulder movement and SD.
Exclusion criteria
- pediatric patients;
- patients without RC tear;
- presence of shoulder pathologies ≥ 2, or previous surgery of the shoulder;
- patients with body surface markers affecting the assessment, such as obesity (body mass index greater than 30);
- patients unable to complete assessment by clinical scores and ROMs.”
We hope this is acceptable.
- Study sample size is missing, which should be discussed within the limitations of the study.
ANSWER: We thank the reviewer for his/her suggestion. In the result section we state:
“A total of 52 patients met the inclusion criteria and were enrolled in this study. The SD assessment identified 32 patients with RC tear and SD (group A) and 20 patients with RC tear without SD (group B). Different SD patterns were stratified within Group A. 4 out of 32 patients were classified as Type 1 SD, 6 as Type 2 SD, and the remaining 22 as Type 3 SD. (Table 1, Group characteristics)”
And in the discussion section we add:
“The strength of this study is the use of a validated clinical method for identifying SD, which showed satisfactory reliability for clinical use [41]. Moreover, the two groups were homogeneous regarding age, weight, height, and tear size. In addition, the use of multiple clinical scales allowed for greater robustness of the results. However, this study has several limitations. First, since this study was single-center and the enrollment time limited, the number of patients included is small and the data should be evaluated in a larger group. A larger group will confirm our findings and can stratify the results achieved in this study by the type of SD (type I, II, III, or IV). Clinical practice and evidence from the literature suggests that SD is a condition that impacts on the scapulohumeral joint kinetics by altering the range of joint motion and negatively impacting the patient’s symptoms. A limitation of the present study is the lack of an analysis on patients’ muscle strength. Subsequent studies are needed to determine whether the loss of strength related to RC tear may be worsened by the co-presence of SD. Finally, due to the lack of validated diagnostic tests to define the presence of SD, it has not been feasible to conduct an analysis to assess whether a longer SD condition would impact the clinical outcomes of patients with RC tear in a statistically significant way.”
We hope it is now acceptable.
- Discussion: the first sentence summarizing the most important findings of the paper is missing
ANSER: We thank the reviewer for his/her suggestion. We added the following state:
“This non-interventional prospective cohort study that analyzed the clinical implications of SD in patients with RC tears showed statistically significant differences in flexion, extension, abduction, and external rotation. On the other hand, no statistically significant difference was identified for internal rotation movements. Statistically significant differences in all outcomes recorded (ASES, CMS, DASH and OSS) were found among the two groups.”
We hope this is acceptable.
- The description of the outcomes (scales) should be reported in the methods.
ANSWER: We thank the reviewer for providing us with the opportunity to improve our work. We now say:
“The study participants completed the Constant and Murley score (CMS) [9], American Shoulder and Elbow Surgeons (ASES) score [5], Disability of the Arm, Shoulder and Hand (DASH) score [18, 19], Oxford Shoulder Score (OSS) [10, 11].
The CMS system [9], normalized for patient age and gender, was used to evaluate preoperative and postoperative shoulder function. It evaluates both subjective and objective function through four domains, including pain (15 points), activities of daily living (20 points), range of movement (40 points), and strength (25 points). The total score ranges from 0, indicating a person with the worst shoulder function, to 100 points, indicating an asymptomatic and healthy person. Pain was assessed with the pain score according to the system of CMS [9], ranging from 0, indicating the severest imaginable pain, to 15 points, indicating no pain.
The ASES score is composed by two sections: the first, defined as pASES, is a patient’s self-report, the second, defined as cASES, is used by physician in order to record the shoulder examination findings. In this study we only considered the patient’s self-report, which is divided in 11 items subcategorised in 2 areas, pain (1 item) and ADL (activity daily living, 10 items). The pain was measured by means of visual analog scale (VAS) divided into 1-cm increments and anchored with verbal descriptors at 0 (not pain at all) and 10 cm (pain as bad as it can be). Every ADL has a score that range from 0 (unable to do) to 3 (not difficult), with a maximum score of 30 [44, 50].
The DASH score [18] is completed subjective score with 30 analogue scale responses and It range from 0 (no disability) to 100 points (severe disability). This standardized questionnaire assesses the symptoms and functional status in people with different upper limb musculoskeletal disorders. It consists of three sections: the first section composed by 30 items evaluates symptoms and functional status at the level of disability; the second and third sections are an optional module of four items for Sport and Music and four items for Work. Each item is scored with a five points scale (1 = no difficulty; 2 = mild difficulty; 3 = moderate difficulty; 4 = severe difficulty; 5 = unable) that are summarized and transformed to define the DASH score.
The OSS was designed as a joint specific instrument to minimise the influence of comorbidity ranging from 12 to 60 points. It is composed by 12 questions and each of them was scored from 1 to 5, with 1 representing best outcome/least symptoms. Scores from each question were added so the best overall score was 12 points. This scoring system was converted to the 0-48 scoring system where the best outcomes is represented by the 37-48 range [10, 11].
The ROMs were measured with a standard universal goniometer. Patients were positioned supine on an examination table with the shoulder abducted 90° in the scapular plane (approximately 15° anterior to the coronal plane). Supine forward elevation (sagittal plane) and internal and external rotation (90° abduction) were scored using standard measurement guidelines. During the test, the examiner (A.B.) stabilized the scapula with one hand while passively assisting the shoulder to reach the position while the forearm was held in neutral rotation. After establishing a firm endpoint, two examiners maintained the shoulder position when a third examiner (U.G.L.) performed the ROM measurement. For each shoulder position, three measurements were taken. Then the average value was determined for statistical purposes [38].”
We hope this is acceptable.
We thank the Editorial Board for having given us the opportunity to revise our manuscript. We appreciate yours and the reviewer’s comments. I hope that the additions have now improved the manuscript, and that it has now reached the standard necessary to be formally accepted for publication in the Journal of Clinical Medicine.
Yours sincerely,
Umile Giuseppe Longo

Round 2
Reviewer 2 Report
The authors replay to all my observations and the manuscript il ameliorated.